# The Comparative Experimental Study of Sodium and Magnesium Dichloroacetate Effects on Pediatric PBT24 and SF8628 Cell Glioblastoma Tumors Using a Chicken Embryo Chorioallantoic Membrane Model and on Cells In Vitro

**DOI:** 10.3390/ijms231810455

**Published:** 2022-09-09

**Authors:** Eligija Damanskienė, Ingrida Balnytė, Angelija Valančiūtė, Vaiva Lesauskaitė, Marta Marija Alonso, Donatas Stakišaitis

**Affiliations:** 1Department of Histology and Embryology, Medical Academy, Lithuanian University of Health Sciences, 44307 Kaunas, Lithuania; 2Laboratory of Molecular Cardiology, Institute of Cardiology, Lithuanian University of Health Sciences, Sukileliu Ave., 50161 Kaunas, Lithuania; 3Clínica Universidad de Navarra, Department of Pediatrics, University of Navarra, 31008 Pamplona, Spain; 4Laboratory of Molecular Oncology, National Cancer Institute, 08660 Vilnius, Lithuania

**Keywords:** pediatric glioblastoma, PBT24, SF8628, sex, dichloroacetate, CAM model, gene expression

## Abstract

In this study, pyruvate dehydrogenase kinase-1 inhibition with dichloroacetate (DCA) was explored as an alternative cancer therapy. The study’s aim was to compare the effectiveness of NaDCA and MgDCA on pediatric glioblastoma PBT24 and SF8628 tumors and cells. The treatment effects were evaluated on xenografts growth on a chicken embryo chorioallantoic membrane. The PCNA, EZH2, p53, survivin expression in tumor, and the *SLC12A2*, *SLC12A5*, *SLC5A8*, *CDH1*, and *CDH2* expression in cells were studied. The tumor groups were: control, cells treated with 10 mM and 5 mM of NaDCA, and 5 mM and 2.5 mM of MgDCA. The cells were also treated with 3 mM DCA. Both the 10 mM DCA preparations significantly reduced PBT24 and SF8624 tumor invasion rates, while 5 mM NaDCA reduced it only in the SF8628 tumors. The 5 mM MgDCA inhibited tumor-associated neoangiogenesis in PBT24; both doses of NaDCA inhibited tumor-associated neoangiogenesis in SF8628. The 10 mM DCA inhibited the expression of markers tested in PBT24 and SF8628 tumors, but the 5 mM DCA affect on their expression depended on the cation. The DCA treatment did not affect the *SLC12A2*, *SLC12A5*, and *SLC5A8* expression in cells but increased *CDH1* expression in SF8628. The tumor response to DCA at different doses indicated that a contrast between NaDCA and MgDCA effectiveness reflects the differences in the tested cells’ biologies.

## 1. Introduction

Despite the latest progress in multimodality treatment, the prognosis of high-grade adult and high-grade pediatric glioblastoma (phGBM) remains dismal [1,2,3]. The nature of glioblastoma is heterogeneous in mitochondrial DNA content, different molecular subtypes, histological differences, and varying degrees of immune cell infiltration [4,5,6,7,8]. Four glioblastoma subtypes have been described using a computational approach, distributed along neurodevelopmental and metabolic axes: proliferative/progenitor, neuronal, mitochondrial, and glycolytic/plurimetabolic. Mitochondrial glioblastoma has been associated with the most favorable clinical outcome; its energy production depends exclusively on oxidative phosphorylation (OXPHOS), whereas the glycolytic/plurimetabolic subtype is supported by aerobic glycolysis, amino acid, and lipid metabolism [2].

The differences between phGBM and adult gliomas, the uniquely heterogeneous developmental origins of pediatric glioblastomas, and the biological factors involved make it clear that therapeutic strategies based solely on adult glioblastomas should be avoided as such an approach does not improve the outcome of phGBM treatment [3,9]. To achieve effective treatment is essential for identifying the individual tumor’s sensitivity to a drug and the mechanisms of its action in carcinogenesis. phGBM cells are more active in glycolysis compared to brain parenchyma cells, and are characterised by an altered metabolism and a response to the availability of extracellular microenvironmental nutrients [10], and can also use alternative substrates such as fatty acids, ketone bodies, and lactate [11]. Their mitochondrial function involves the OXPHOS activation of HpGBM cells [12], promotes the biosynthesis of DNA, RNA, lipids, and proteins, and maintains the rapid proliferation of cells [13]. OXPHOS is also potentially important for resistance to chemotherapy and radiation [12,14]. Targeting the pathophysiological metabolic pathways of hpGBM could be used as a new approach to improve tumor treatment [15]. Inhibition of pyruvate dehydrogenase (PDH) by pyruvate dehydrogenase kinase (PDK) activity is associated with phGBM pathophysiology. One of the promising targets is the PDK-PDH axis [16].

Researchers have reported that boys with phGBM have significantly shorter overall and progression-free survival than girls [17]. Glycolytic metabolites selectively discriminate between males. A sex-linked dimorphism in pyruvate metabolism has also been identified, suggesting a potential synergy among sex, glioma metabolism, and genomic variations in a glioma patient’s outcome [18]. Elucidating the gender-related differences in the pharmacology of DCA, mechanisms of action, and related treatment efficacy may be necessary for personalised treatment approaches to reveal sex-related differences in effectiveness and safety [19].

We have previously reported the differences in the effects of dichloroacetate (DCA) preparations in treating female pediatric and adult high-grade glioblastoma cell tumors transplanted on a CAM [20]. The present study’s novelty is that it presents the effects of DCA on the mechanisms of tumorigenesis of pediatric glioblastoma PBT24 and SF8628 cell tumors in vivo (using the CAM model) and in cells in vitro to determine the relevance of DCA in terms of its efficacy for personalized phGBM therapy. Understanding the mechanisms of drug action is also particularly important when it comes to the use of drug-preparation combinations in treatment. 

As an active substance, DCA has been tested as an investigational medicine for treating lactic acidosis caused by congenital mitochondrial disorders [19,21]. In vitro and animal studies have shown that DCA can act as a mitochondrial metabolic regulator of cancer cells and alter the glycolytic cell phenotype [22,23]. DCA, by inhibiting the E1α subunit of pyruvate dehydrogenase kinase-1 (PDK1), keeps the PDH complex in its unphosphorylated active form, facilitates the oxidative removal of pyruvate and the mitochondrial oxidation of glucose, and decreases the amount of lactic acid in cancer cells and their microenvironment [24,25]. The formation of intracellular lactate produces NAD^+^, which supports glycolysis. Extracellular lactate provides an acidic microenvironment important for tumor invasion [26] and tumor adaptation under hypoxic conditions [26,27], and it also suppresses the immune system [27]. PDH is regulated by the reversible phosphorylation of PDK1 [28]. DCA treatment activates PDH, while pyruvate targets acetyl-CoA production and causes OXPHOS via the activation of mitochondrial complex I [29]. DCA treatment increases ROS production and promotes proapoptotic changes in cancer cells [30].

DCA is transported into the cell by a monocarboxylate transporter (SLC5A8) and enters the mitochondrial matrix via mitochondrial pyruvate carriers. When administered per os, it is rapidly resorbed via the intestine. DCA crosses the blood–brain barrier and its concentration in the cerebrospinal fluid can be measured. After per os administration of DCA, blood lactate concentrations start to decrease in approximately 15–30 min and this can be used to evaluate the effect on PDH [28].

This study investigated tumors formed from a high-grade glioblastoma SF8628 (3-year-old girl’s) and a high-grade glioblastoma PBT24 (13-year-old boy’s) cell line cells using a CAM model; we tested the tumor response to treatment with magnesium or sodium dichloroacetate salt preparations (MgDCA and NaDCA) on PCNA, EZH2, p53, and survivin expression in xenograft tumors on CAM; and we explored the effects of MgDCA and NaDCA on Na-K-2Cl (*SLC12A2*), K-Cl (*SLC12A5*), SLC5A8 co-transporters, and E-cadherin (*CDH1*) and N-cadherin (*CDH2*) gene expression in studied phGBM cells.

## 2. Results

### 2.1. Stereomicroscopic Findings of Transplanted PBT24 and SF8628 Cell Tumors on CAM

Figure 1 shows the PBT24 tumor in the control group was demarcated at EDD9 and blurred at EDD12 with two zones of different intensity: the tumor size appeared smaller at EDD12 due to tumor invasion into the mesenchyme; the tumor on the surface of the CAM was more intensely coloured, with a distinct vascular network (“spoked-wheel”) formed around the tumor (EDD12, and in CAM ex ovo). Compared to the PBT24-control, the tumors treated with 10 mM NaDCA and 5 mM MgDCA in the PBT24 showed a blunted tumor growth, with a less pronounced vascular network surrounding the tumor. Clear tumor outlines indicated tumor growth on the CAM surface (Figure 1, H–E).

The stereomicroscopic EDD12 SF8628 control tumor was larger than the 10 mM NaDCA-treated tumor. The histological H–E image shows the invasive growth pattern of the control tumor. The EDD12 tumor treated with 10 mM NaDCA showed little change in size compared to the EDD9 tumor (tumor growth on the CAM surface). The SF8628-5 mM MgDCA also grew on the CAM surface. Comparison of the H–E histological images of EDD12 in SF8628-10 mM NaDCA and SF8628-5 mM MgDCA showed that the tumor treated with 5 mM MgDCA tissue was denser than that of the one treated with 10 mM NaDCA. 

Figure 2 shows the vascular network around the tumor, highlighted by the injection of fluorescent dextran into the CAM vessel at EDD12; the “spoked-wheel” was well defined around the PBT24-control tumor. Compared to the control, neoangiogenesis was weaker around the PBT24-5 mM MgDCA and the PBT24-10 mM NaDCA tumors. Around the SF8628-control tumor, EDD12 showed a “spoked-wheel” type vascularisation. The vascular network around the SF8628-control tumor was denser than around the 10 mM NaDCA- and 5 mM MgDCA-treated tumors. The dextran green fluorescence was the most intense in the control tumors. This indicated that the blood vessels from the CAM mesenchyme penetrated and vascularised the tumor. In the treated tumors, the green fluorescence intensity was pronounced at the tumor margins, indicating that vascular ingrowth into the tumor was blunted, i.e., suppressed neoangiogenesis. The attenuated neoangiogenesis is also indicated by the yellow fluorescent colour of the treated tumor (Figure 1: EDD12, CAM ex ovo; Figure 2).

### 2.2. PBT24 and SF8628 Tumor Growth, Tumor Invasion into CAM Rate, the Number of Blood Vessels, and CAM Thickness

Compared to the PBT2 control, the number of invasive tumors was significantly reduced in the groups treated with 10 mM NaDCA and 5 mM MgDCA PBT24. The PBT24 tumors treated with 10 mM NaDCA had a significantly lower invasion rate into the CAM than those treated with 5 mM NaDCA. The 5 mM MgDCA dose effect on invasion was higher than 5 mM NaDCA. Among the NaDCA-treated PBT24 groups, the invasion rate differed depending on the drug concentration; 10 mM NaDCA had a better effect. Compared with the PBT24-control group, treatment with 2.5 mM MgDCA had no significant (*p* > 0.05) effect on tumor invasion (Table 1; Figure 3).

Compared to the SF8628-control, all doses of DCA except 2.5 mM MgDCA significantly reduced the incidence of tumor invasion into the CAM; the strongest inhibitory effect on invasion was observed in the 10 mM NaDCA-treated group. The incidence of invasion was significantly lower in the SF8628 tumors treated with 10 mM NaDCA than in the tumors treated with 2.5 mM MgDCA (Table 1; Figure 3).

Compared to the PBT24-control, the sub-tumor CAM thickness was significantly lower in the PBT24 tumors treated with 2.5 mM MgDCA and 5 mM MgDCA. Treatment with 10 mM and 5 mM NaDCA had no significant effect on the CAM thickness under the PBT24 tumor (Table 1). Treatment with NaDCA and MgDCA doses did not significantly impact the CAM thickness under the SF8628 tumor. The strongest effect on CAM thickness was observed in the 5 mM MgDCA-treated group (Table 1).

The number of CAM vessels under the tumor is shown in Table 1. When comparing the PBT24 tumor groups, the dose of 5 mM MgDCA had the significantly strongest inhibitory effect on neoangiogenesis. Compared to the PBT24-control, no significant changes in the number of blood vessels were found in the other treated groups. Comparing the number of blood vessels in the CAM mesenchyme under tumors in the SF8628-control group with the groups treated with 10 mM NaDCA and 5 mM NaDCA, we found that both doses of NaDCA significantly reduced the number of blood vessels in the membrane. The number of vessels under the tumors in the CAM of the 5 mM and 2.5 mM MgDCA-treated groups was not significantly different from that of the SF8628-control. Still, the number of vessels in both of the MgDCA-treated groups was significantly higher than in the group treated with 10 mM NaDCA. The dose of 10 mM NaDCA had the most inhibitory effect on neoangiogenesis (Table 1).

### 2.3. The PCNA Expression of PBT24 and SF8628 in Control and DCA-Treated Tumors

Table 2 and Figure 4 show the expression data of PCNA-positive cells in the PBT24 and SF8628 tumor groups tested. The number of PCNA-positive cells in the PBT24-control tumor accounted for 63.8% of the total number of cells, while in the SF8628-control tumor, the number of PCNA-positive cells accounted for 90.8%. PCNA expression in the PBT24-control tumor was significantly lower than in the SF8628-control. The effects of the doses of NaDCA and MgDCA treatment on PCNA expression are presented in Table 2 and Figure 4.

Compared to the PBT24-control, all treatments except 2.5 mM MgDCA reduced the PCNA-positive cell number in the PBT24 tumors. Both doses of NaDCA effectively reduced PCNA expression, but no difference in effect was found when comparing the two NaDCA-treated groups. When comparing the PBT24 groups treated with different doses of MgDCA, the 5 mM MgDCA dose was significantly more effective than the 2.5 mM MgDCA dose. Both doses of NaDCA significantly reduced PCNA expression in the tumors to a greater extent than PBT24-2.5 mM MgDCA (Table 2).

Compared to the SF8628-control, all investigational preparations significantly reduced the number of PCNA-positive cells in the SF8682 tumor tissue. The strongest inhibition of PCNA expression was found at doses of 5 mM MgDCA and 10 mM NaDCA; no difference was found when comparing these groups. The 10 mM dichloroacetate anion dose had a significantly stronger effect than the corresponding 5 mM anion dose, independent of the cation. Still, when comparing the effect of 2.5 mM MgDCA with 5 mM NaDCA, a significantly better impact on PCNA expression was found for the 5 mM NaDCA. After DCA treatment, the expression of PCNA-positive cells in the SF8628 tumors remained higher than in the PBT24 tumors of the respective groups, except in the tumor treated with the 5 mM MgDCA dose (Table 2; Figure 4).

### 2.4. EZH2 Expression of PBT24 and SF8628 in Control and DCA-Treated Tumors

Table 3 and Figure 5 show the expression data of EZH2-positive cells in the PBT24 and SF8628 tumor groups tested. The number of EZH2-positive cells in PBT24-control tumors was 71.0% of the total number of cells; in SF8628-control tumors it was 85.4%. EZH2 expression was lower in the PBT24-control than in the SF8628-control tumors (*p* < 0.02).

Compared to the PBT24-control, all the DCA-treated groups, except 2.5 mM MgDCA, reduced the number of EZH2-positive cells in the PBT24 tumors. No significant difference was found when comparing the effects of both NaDCA doses (*p* > 0.05). The expression of EZH2 was significantly lower in the PBT24-5 mM MgDCA group tumors than in those treated with 2.5 mM MgDCA. The tumors treated with both doses of NaDCA were more effective than those treated with 2.5 mM MgDCA. EZH2 expression was significantly more strongly inhibited by the 5 mM MgDCA dose than by the 2.5 mM MgDCA dose (Table 3; Figure 5).

Compared to the SF8628-control, all of the preparations significantly reduced the number of EZH2-positive cells in the SF8682 tumors. EZH2 expression was significantly lower in those treated with 10 mM NaDCA than those treated with 5 mM NaDCA. No significant difference was found between the tumors treated with 5 mM and 2.5 mM MgDCA (*p* > 0.05). As in the PBT24 tumors, EZH2 expression was strongly inhibited by the 5 mM MgDCA dose (Table 3; Figure 5). In the SF8628-control tumors, the correlation between EZH2 and p53 was 1.0 (*p* = 0.017); in the PBT24 control tumors, this correlation was 0.4 (*p* > 0.05).

### 2.5. The p53 Expression in Control and DCA-Treated PBT24 and SF8628 Tumors

Table 4 and Figure 6 show the expression data of p53-positive cells in the PBT24 and SF8628 tumors tested. The p53-positive cell count was 75.1% in the PBT24-control tumors and 85.3% in the SF8628-controls. No difference in p53 expression was found when comparing the control groups with each other (*p* > 0.05).

Compared to the PBT24-controls, the tested DCAs reduced the number of p53-positive cells in the PBT24 tumors. The strongest inhibition of p53 expression was observed with 10 mM dichloroacetate (10 mM NaDCA and 5 mM MgDCA), and no difference was found between these groups. In the PBT24-10 mM NaDCA group, tumor p53 expression was significantly lower than in those treated with 5 mM NaDCA. The PBT24-10 mM NaDCA tumor had significantly lower p53 expression than the 2.5 mM MgDCA tumor. A comparison of p53 expression between the PBT24-5 mM MgDCA and PBT24-2.5 mM MgDCA tumors showed no difference in the expression of p53 (*p* > 0.05; Table 4; Figure 6).

Compared to the SF8628-control, all the tested preparations except 2.5 mM MgDCA significantly reduced the number of p53-positive cells in the SF8682 tumor. No significant difference in the p53 expression was found between groups treated with different concentrations of NaDCA (*p* > 0.05). Similarly, there was no difference in the p-53 expression when comparing the groups treated with different MgDCA doses (*p* > 0.05). In the SF8628-10 mM NaDCA tumor, the p53 expression was significantly lower than in the tumor treated with 2.5 mM MgDCA (Table 4; Figure 6).

### 2.6. The Survivin Expression in the Control and DCA-Treated PBT24 and SF8628 Tumors

Table 5 and Figure 7 show the expression data of survivin-positive cells in the PBT24 and SF8628 tumor groups. The survivin-positive cell count was 53.8% in the PBT24 and 52.8% in the SF8628 controls.

Compared to the PBT24-control, only the 10 mM NaDCA dose significantly reduced the number of survivin-positive cells in the PBT24 tumor; this dose was more effective in reducing the expression of the marker than the treatment with 5 mM NaDCA and 2.5 mM MgDCA. No difference in survivin expression between the tumors treated with 5 mM and 2.5 mM MgDCA was found (*p* > 0.05).

A comparison of the survivin expression among SF8628-control group and the tumors treated with 10 mM NaDCA and 5 mM MgDCA revealed that the marker’s expression was significantly lower in the treated tumors than that of the control. The expression of the marker was reduced by the 10 mM NaDCA but not by the 5 mM NaDCA. A comparison of survivin expression in the tumors treated with different doses of MgDCA showed no significant difference in expression (Table 5; Figure 7).

### 2.7. The Expression of NKCC1, KCC2, SLC5A8, E- and N-Cadherins Genes in the Studied PBT24 and SF8628 Cells Groups

The expression data of *SLC12A2*, *SLC12A5*, *SLC5A8*, and *GAPDH* in the studied PBT24 and SF8628 cells are shown in Table 6.

A 3 mM dichloroacetate anion concentration of NaDCA or MgDCA preparations, regardless of the cation present in the salt, did not effect the *SLC12A2* expression in the PBT24 and SF8628 cells (Table 6). 

There was no significant effect of 3 mM dichloroacetate (NaDCA or MgDCA preparations) on *SLC12A5* expression in the PBT24 and SF8628 cells. The expression of *SLC12A5* in the SF8628-control and SF8628 cells treated with NaDCA or MgDCA was significantly lower than in the corresponding PBT24 cell groups (Table 6). 

No significant effect of 3 mM dichloroacetate (NaDCA or MgDCA preparations) on the *SLC5A8* expression in the PBT24 and SF8628 cells was found compared with the respective control. The expression of *SLC5A8* in the PBT24-control and NaDCA- and MgDCA-treated cells did not differ from that observed in the corresponding groups of SF8628 cells (Table 6). 

The *CDH1*, *CDH2*, and *GAPDH* expression data of the studied PBT24 and SF8628 cell groups are shown in Table 7 and Figure 8.

Treatment with both salts of DCA did not affect the *CDH1* expression of PBT24 cells. Compared to the SF8628-control, the SF8628-1.5 mM MgDCA group showed a significantly higher *CDH1* expression, i.e., the treatment increased the expression. The SF8628-1.5 mM MgDCA cells showed significantly higher *CDH1* expression than the cells treated 3 mM NaDCA. The SF8628-control and SF8628 cells treated with NaDCA or MgDCA were significantly lower in *CDH1* expression than the corresponding PBT24 cell groups (Table 7; Figure 8). 

No effect of 3 mM dichloroacetate (NaDCA or MgDCA preparations) on the *CDH2* expression of the PBT24 and SF8628 cells was found compared with the respective controls. The PBT24-1.5 mM MgDCA group showed a significantly higher *CDH2* expression than the corresponding group of SF8628 cells (Table 7).

## 3. Discussion

phGBM is the most malignant brain tumor with an unfavorable prognosis. A major challenge in treating brain tumors is developing drugs that cross the blood–brain barrier and enter the tumor cells. The importance of this challenge is even more significant when surgical treatment is not feasible in the case of children and is rendered unresectable [31]. In such cases, in addition to radiotherapy, effective chemotherapy remains essential. The WHO Tumors of the Central Nervous System Classification (CNS5) demonstrates the heterogeneity of gliomas and their different types, reflecting the other histopathological characteristics and molecular-genetic profiles of phGBM [32,33]. Researchers have suggested that cellular oxidative metabolism is a factor in the etiology of phGBM and that the development of these tumors is associated with possible mitochondrial and nuclear-mitochondrial DNA polymorphisms in tumor genesis [34].

In cells with mitochondrial defects, DCA inhibits PDK and indirectly activates PDH by inhibiting PDK-mediated PDH phosphorylation [15]. The ability of DCA to abrogate cancer cells‘ resistance to apoptosis and increase their sensitivity to anticancer agents makes DCA promising for combination with chemo- or radiotherapy [8,35,36].

Recently the different temozolomide and valproic acid effects on tumorigenesis mechanisms have been reported, such as their effect on tumor growth, the histological expression of PCNA and EZH2 in tumor cells, Na-K-2Cl, K-Cl, and SLC5A8 co-transporter gene expression in pediatric glioblastoma PBT24 and SF8628 cell tumors on chick embryo chorioallantoic membranes (CAM) and on corresponding cells in vitro, highlighting the importance of studies on efficacy in the context of individualized anticancer therapy [37,38]. This study’s novelty lies in the differential effect of the active substance DCA‘s anions on PBT24 and SF8628 tumors on the CAM and the cells in vitro, comparing the differences in the impact of the studied NaDCA and MgDCA salts. The observations suggested that Mg^2+^ transport and Mg^2+^ binding were enhanced in glioma cells and provided a strong rationale for the further investigation of glioblastoma due to altered Mg^2+^ homeostasis and cation transport as potential therapeutic targets [39]. A sufficient concentration of mitochondrial Mg^2+^ is necessary to maintain the electron transfer chain integrity and to control the flux of metabolites during the TCA cycle and glycolysis [40]. Mg^2+^ in mitochondria is a significant contributor to a cell’s resistance to oxidation, i.e., its ability to repair or avoid oxidative damage [41].

The CAM model has been used for decades as a model for in vivo cancer research to test experimental hypotheses in pre-clinical studies [42]. By embryonic day 10, a chick embryo is considered an immunodeficient model, allowing human cells to be transplanted on the CAM [43]. The CAM is a convenient model for drug efficacy in tumor growth and neoangiogenesis studies [44,45]. In the CAM model, we found no differences between the PBT24 and SF8628 controls in the rate of tumor invasion into the CAM, the number of blood vessels in the CAM under the tumors and the thickness of the CAM. Treatment with DCA showed that these tumors differed in their sensitivity to treatment, depending not only on the dose of DCA but also on the cation present in the salt. The PBT24 and SF8628 invasive tumor rates were equally reduced by treatment with 10 mM NaDCA or 5 mM MgDCA, while treatment with 5 mM NaDCA significantly reduced only the SF8628 tumor invasion frequency.

CAM mesenchyme thickening is an added criterion for evaluating tumor xenografts’ malignancy [20,46]. Contrary to NaDCA, the MgDCA doses significantly reduced the CAM thickness under the PBT24 tumor, while the SF8628 treatment with the tested doses of both DCA salts did not effect the CAM thickness. Tumor xenografts induce an inflammatory response in CAM mesenchymes, accompanied by neoangiogenesis and an increased membrane mesenchyme exfoliation [47,48,49]. PBS solution does not cause membrane thickening, but hyperosmolar solutions enable it [50]. Other researchers have reported that NaDCA inhibited angiogenesis [51]; the effect of a 10 mM of DCA anion concentration on CAM thickening is dependent on salt‘s cation [20]. This study shows that MgDCA and NaDCA had a different impact on neo-angiogenesis in the PBT24 and SF8628 tumors; only the 5 mM MgDCA dose inhibited PBT24 tumor-related neoangiogenesis, while only both the NaDCA tested doses inhibited the development of blood vessels in the CAM under the SF8628 tumor. The effect of the tested substances on tumor growth, CAM thickness, and neoangiogenesis showed a trend towards SF8628 being more sensitive to NaDCA than PBT24. At the same time, the latter was more sensitive to MgDCA.

The PBT24 control tumor had lower PCNA and EZH2 expression than the SF8628 tumor, while p53 and survivin expression did not differ between the tested tumors on the CAM; the 10 mM of DCA-containing preparations significantly reduced EZH2 expression in the SF8628 and PBT24 tumors, abolishing this difference between the PBT24 and SF8628 tumors. In contrast to the PBT24 tumor, the SF8628 tumor’s PCNA and EZH2 expressions were sensitive to 2.5 mM MgDCA. However, comparing the effect of the 5 mM DCA preparations, the impact of NaDCA on PCNA expression was significantly better in the PBT24 tumor. PCNA expression is associated with poor survival and an advanced glioblastoma stage and is recognized as a diagnostic and prognostic biomarker and an effective target for tumor treatment [52]. In glioblastomas with increased EZH2 expression, a prolonged depletion of EZH2 leads to a change in cell survival [53], suggesting that it is worthwhile to investigate the efficacy of treating tumors with EZH2 inhibitors. EZH2 is currently thought to play a pro-tumorigenic role in pediatric gliomas, specifically in diffuse midline gliomas (DMG), where a mutated histone H3K27M is present in 80% of cases, and pharmacological inhibition of EZH2 activity by EZH2 inhibitors results in the inhibition of proliferation in DMG cells [54,55]. Recent research has shown that EZH2 is a potential target for DMG [55,56]. The FDA has approved the EZH2 inhibitor tazemetostat for the therapeutic indication of epithelioid sarcoma and follicular lymphoma; thus, there are safe EZH2 inhibitors available. Therefore, pre-clinical data are encouraging clinical trials [57]. Immunohistochemical analysis of samples from DMG patients has shown a positive correlation between Ki-67 and EZH2; the trend of this correlation was strong, if not statistically significant, due to the tiny patient cohort [58,59]. EZH2 inhibition decreased glioma cell proliferation in vitro and increased the survival of mice with H3F3AK27M-mutated gliomas [55].

On the other hand, experimental studies have suggested that EZH2 may inhibit pediatric glioblastomas. Studies of DMG mouse models have indicated that EZH2 plays a tumor-suppressing role. This was demonstrated by the discovery of transcriptional programs resulting from the genetic disruption of EZH2, i.e., an increased interferon-γ response in the loss/gain-of-function DMG model and an increased oxidative phosphorylation/mitochondrial metabolic signature in the gain-of-function model [58]. Such data suggest that EZH2 inhibitors in pediatric glioblastoma treatment should be investigated further [58]. We recently reported that TMZ did not alter EZH2 expression but significantly reduced PCNA expression in PBT24 and SF8628 tumors on CAM [37]. In the SF8628 control tumor, the correlation between EZH2 and p53 was solid and significant, while in the PBT24 control tumor, the correlation was weak and insignificant.

The gene of p53 is essential for carcinogenesis. Reduction/inhibition of p53 is thought to be beneficial due to its mutated/oncogenic type expression. p53 expression correlates with unfavorable outcomes and therefore is a target in the treatment of glioblastoma [60]. The pathobiology of the spontaneous malignant proliferation of tumor cells requires new anti-tumor non-cytotoxic drugs to resume cell maturity and combine drugs to inhibit p53/p16. This is a common therapeutic practice and an integrated approach to treating glioma [57,61]. The most potent inhibition of p53 expression was found with 10 mM dichloroacetate preparations (10 mM NaDCA and 5 mM MgDCA), with no difference among the PBT24 and SF8628 tumors. The 2.5 mM MgDCA preparation was not effective on p53 expression in both studied tumors.

This study showed the differences in the effect of DCA on survivin expression in the tumors, which is related to the cation. Only 10 mM NaDCA reduced PBT24 tumor survivin expression. The 10 mM NaDCA and 5 mM MgDCA doses showed a marked decrease in survivin expression in the SF8628 tumor on the CAM. The survivin protein is a member of the apoptosis inhibitor family, which regulates cell division and inhibits apoptotic cell death [62] and frequently is expressed in various cancer cells [63,64]. Survivin expression in gliomas is related to poor prognosis [65,66,67] and resistance to chemotherapy [68].

DCA transport into the cell by monocarboxylate cotransporter SLC5A8 depends on Na^+^ and Cl^−^ ions; DCA induces cell apoptosis via pyruvate-dependent HDAC inhibition [69]. It is important to determine the drug effect on the [Cl^−^]i regulation. High-grade glioblastoma cells are characterised by elevated levels of [Cl^−^]i [70], which is due to increased *SLC12A2* and decreased *SLC12A5* co-transporter activity [71,72]. Cl^−^ is essential for cell volume and apoptosis mechanisms [73]. In human glioblastoma cells, increased SLC12A2 protein expression is associated with tumor grade and cell migration. Inhibition of SLC12A2 suppresses glioblastoma cell invasion [70,74,75]. This study showed that the *SLC12A2* and *SLC5A8* expression in the PBT24-control and PBT24 DCA-treated cells did not differ from its expression in the corresponding SF8628 cell groups. The treatment with the studied DCA preparations did not effect the expression of *SLC12A2*, *SLC12A5*, and *SLC5A8* in the PBT24 and SF8628 cells. The DCA did not increase SLC12A2 expression, suggesting that it could be used in combination with other anticancer agents, such as temozolomide, which increases SLC12A2 expression in phGBM cells [37]. *SLC12A5* expression in the SF8628 control and DCA-treated cells was significantly lower than in the PBT24 cells. An early sign of apoptosis is a reduction in cell volume due to intracellular K^+^ and Cl^−^ loss [76,77], so the higher expression of *SLC12A5* in the PBT24 cells may also be linked to better apoptotic features.

The control SF8628 and DCA-treated cells in the study showed a significantly lower *CDH1* expression than the corresponding PBT24 cell groups. The dose of 1.5 mM MgDCA significantly increased *CDH1* expression in the SF8628 cells (MgDCA-treated cells showed a considerably higher *CDH1* expression than NaDCA-treated cells), while in contrast, the DCA preparations did not effect *CDH1* expression in the PBT24 cells. The DCA preparations did not effect *CDH2* expression in the PBT24 and SF8628 cells. The role of *CDH1* as a tumor suppressor has previously been documented in cancer [78]. The loss of *CDH1* function, gene expression, or suppression resulted in mesenchymal morphology and increased cell migration and invasion in certain epithelial cancers [79]. 

On the other hand, reintroducing CDH1 into non-expressing cells changes the phenotype of poorly differentiated cancers to a well-differentiated, minimal invasive epithelioid type with well-established cell junctions [79,80,81]. A decrease in CDH1 expression is linked to astrocytoma progression [82,83]. However, there is some evidence that high CDH1 expression is related to a worse prognosis [84]; thus, the contribution of *CDH1* to cancer progression needs further investigation [81].

The differences found between the PBT24 and SF8628 cells may be sex-related. Further studies analysing the effect of sex on the efficacy of the drug in determining the molecular subtype of glioblastoma may lead to individualising treatment according to gender-specific differences in molecular mechanisms [85]. The potential sex-related effects of DCA have been recently reviewed [19]. Adverse effects of DCA treatment in adults usually manifest as reversible peripheral neuropathy. At the same time, DCA can safely be used in children with inherited mitochondrial disorders for several years [21], suggesting that DCA may be safe for pediatric patients.

A limitation of this study is that it covered only two tumor cell lines. It is essential to perform studies with a more significant number of cell lines for individual therapy issues. Another important fact is that cell lines do not always accurately reflect the characteristics of primary cells. However, the data from this study provided an essential basis for further research. It is, therefore, necessary to continue this study with primary cells obtained after GBM surgery.

## 4. Materials and Methods

### 4.1. Cell Lines and Cell Culture

Pediatric PBT24 cells (donated by Prof. M.M. Alonso; University of Navarra, Spain) [86] and diffuse intrinsic pontine glioblastoma SF8628 cells harboring the histone H3.3 Lys 27-to-methionine (Sigma Aldrich, St. Louis, MO, USA) [87,88] were studied. The PBT24 cells were cultivated in 1640 RPMI medium (Sigma Aldrich, St. Louis, MO, USA), and the SF8628 cells were grown in DMEM–High-Glucose medium (Sigma Aldrich, St. Louis, MO, USA). The media were supplemented with 2 mM L-Glutamine (Sigma Aldrich, St. Louis, MO, USA), 10% fetal bovine serum (Sigma Aldrich, St. Louis, MO, USA), 100 µg/mL of streptomycin and 100 IU/mL of penicillin (P/S; Sigma Aldrich, St. Louis, MO, USA). Cells were incubated in a humidified 5% CO_2_ atmosphere at 37 °C temperature.

### 4.2. Groups Studied for PBT24 and SF8628 Tumors on CAM

The study groups of PBT24 and SF8628 tumors were evaluated for tumor growth; their invasion, the number of blood vessels in the CAM beneath the tumor, the thickness of the CAM, and the immunohistochemical marker studies and the group sample are listed in Table 8.

The doses of NaDCA and MgDCA chosen to study the efficacy of DCA on GB tumor in vivo are based on our previous study [20].

### 4.3. Application of the CAM Model to the Study of PBT24 and SF8626 Tumors on CAM

According to Lithuanian and EU legislation, research using the CAM model does not require the Ethics Committee‘s approval. Fertilized eggs of Cobb 500 hens were obtained from a hatchery (Rumšiškės, Lithuania). In vivo xenograft biomicroscopy during embryonic development days 9–12 (EDD9–12) is suitable for assessing tumor growth characteristics and malignancy and determining tumor sensitivity to treatment. The formed tumors were transplanted onto the CAM on embryonic day 7 (EDD7). Stereomicroscopy of the tumors was performed on days 2 and 5 post-transplantation (i.e., EDD9–12). On day 5 of tumor growth (EDD12), the tumors with CAM were resected and fixed at 10% formalin. Histological specimens were stained with hematoxylin and eosin (H–E) and immunohistochemical (IHC) techniques. Preparation of fertilized eggs for the assay, tumor formation from the test cells, tumor transplantation onto the CAM in vivo, EDD time course biomicroscopy to assess the effect of the drug on tumor growth, neo-angiogenesis, and the histological examination of the tumor, assessment of the number of blood vessels in the CAM beneath the tumor, and the thickness of the CAM were performed in the same method as we have reported previously [37].

The H–E-stained samples were divided into non-invasive and invasive. The non-invasive tumors were located on the surface of the CAM and the integrity of chorionic epithelial (ChE) cells were not damaged. Destruction of ChE integrity or invasion of tumor cells into the mesenchyme of the CAM was classified as the tumor invasion into CAM type. Tumor invasion was assessed in H–E specimens at 20× and 40× magnification.

### 4.4. Immunohistochemical Study 

The expression of EZH2 and PCNA in tumor cells was measured by immunohistochemistry as described by us previously [37]. IHC analysis of p53 (aa 211-220, clone240, CBL404, Merck, Germany) and survivin expression was performed on tumors fixed for 24 h at 22 °C temperature with 10% formalin and paraffin-embedded using standard procedures. Thin CAM with tumor sections of 3 µm was mounted onto adhesion slides (P0425-72EA, Sigma-Aldrich, St. Louis, MO, USA), dried for 24 h at 37 °C temperature, deparaffinized, and rehydrated by standard techniques. A Tris/EDTA buffer pH 9 (DM828, Agilent Dako, Glostrup, Denmark) was used for heat-induced antigen retrieval in a decloaking chamber (DC12-220-0121, Biocare Medical, CA, USA) at 95 °C for 20 min. The Shandon CoverPlate System (72110013, Epredia, Machelen, Belgium) was used for staining. Peroxidase blocking reagent (S2023, Agilent Dako, Glostrup, Denmark) was used to block endogenous peroxidase.

Slide treatment with the primary survivin monoclonal antibody (1H5, MA5-17035, Invitrogen, CA, USA) diluted in Antibody Diluent with Background Reducing Components (DM830, Agilent Dako, Glostrup, Denmark) at a 1 µg/1 mL concentration for 12 h at 4 °C temperature was performed. The primary antibody–antigen complex was detected using horseradish peroxidase-labeled polymer dextran conjugated to a secondary mouse antibody, with a linker (SM802 and SM804, respectively; Agilent Dako, Glostrup, Denmark) at room temperature for 30 min. Immunostaining was visualized using 3,3′-diaminobenzidine-containing chromogen (DAB, DM827, Agilent Dako, Denmark) diluted with a substrate buffer (SM803, Agilent Dako, Glostrup, Denmark); the staining was visible at the site of the target antigen, which was recognized by the primary antibody as a brown color. A tris-buffered saline solution containing Tween 20 (DM831, Agilent Dako, Glostrup, Denmark) wash buffer was used. Slides were counterstained with a hematoxylin solution (SM806, Agilent Dako, Glostrup, Denmark).

To assess of the studied marker’s protein expression, 2 random vision fields (plot area: 23,863.74 µm^2^) of the IHC-stained tumor were photographed at 40× magnification. All cells and the marker-positively stained cells were computed, and the percentage of marker-positive cells was counted in the tumor.

### 4.5. Extraction of RNA from Study Cells and Determination of the Gene Expression in PBT24 and SF8628 Cells

PBT24 and SF8628 cells were treated with 3 mM NaDCA and 1.5 mM MgDCA for 24 h. The DCA concentrations were selected according to other researchers’ data [36,89]. Tested cell groups were grown in a cell-culture medium depending on the cell line. RNA expression assays were performed for *GAPDH* (Hs02786624_g1), *SLC5A8* (Hs00377618_m1), *SLC12A2* (Hs00169032_m1), *SLC12A5* (Hs00221168_m1), *CDH1* (Hs01023894_m1), and *CDH2* (Hs00983056_m1) genes. The total RNA were extracted, and a reverse transcription and real-time polymerase chain reaction (PCR) were conducted according to the manufacturer’s instructions, as reported by us [37]. The expression of *SLC12A5*, *SLC12A2*, *SLC5A8*, *CDH1*, and *CDH2* was investigated in the controls and the groups treated with 3 mM NaDCA and 1.5 mM MgDCA for 24 h (*n* = 6 per group).

### 4.6. Statistical Analysis

The IBM SPSS Statistics V23.0 statistical package was used to perform the statistical analysis. The tumor invasion frequency into the CAM was represented as a percentage. The χ2 test was used to compare the incidence of tumor invasion frequency between the groups. The assumption of normality was tested using the Shapiro–Wilk test. The marker-positively stained cells data, the blood vessels number, and the thickness of the CAM were described as median and range (min and max) values. A difference between the two independent groups was assessed using the Mann–Whitney U test.

To investigate the expression of *SLC12A5*, *SLC12A2*, *SLC5A8, CDH1*, and *CDH2* in the control and 3 mM NaDCA- and 1.5 mM MgDCA-treated groups, the CT (threshold cycle) value was normalized with *GAPDH* to obtain a ΔCT value. The ∆∆CT (Livak method) was used to calculate the relative fold change in gene expression level [90]. The difference at the *p* < 0.05 value was considered significant. To create figures, the GraphPad Prism 7 software was used.

## 5. Conclusions

The differences between the PBT24 and SF8628 control tumors in the PCNA, EZH2, and EZH2–p53 correlation expression in the tumor on the CAM, and the *SLC12A5* expression in the studied cells suggested a possible distinct malignancy of the investigated pediatric glioblastoma cells. The disparities in the effect of the DCA-tested preparations on tumor growth indicated that the efficacy of DCA may be dependent not only on the dose of the preparation but also on Na^+^ or Mg^2+^ cations. On the other hand, these properties may reflect the differences in the cell tumorigenesis pathway regulation of SF8628 and PBT24 cells.

## Figures and Tables

**Figure 1 ijms-23-10455-f001:**
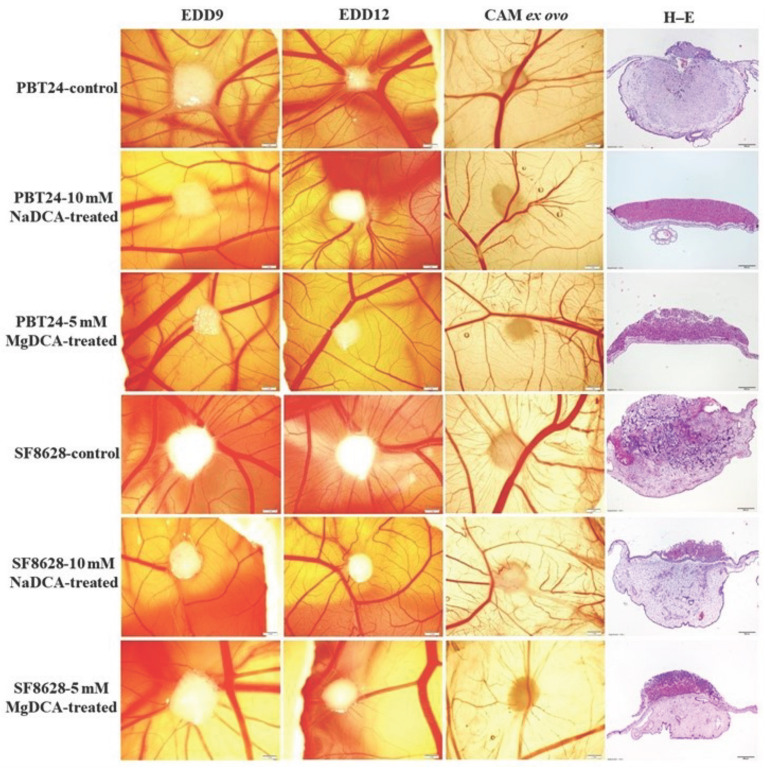
Stereomicroscopic and histologic images of PBT24 and SF8628 tumors in vivo on the CAM, the tumor ex ovo, and the H–E stained tumors histologically. EDD9, EDD12, and CAM ex ovo scale bar is 1 mm; H–E preparations’ scale bar is 200 µm.

**Figure 2 ijms-23-10455-f002:**
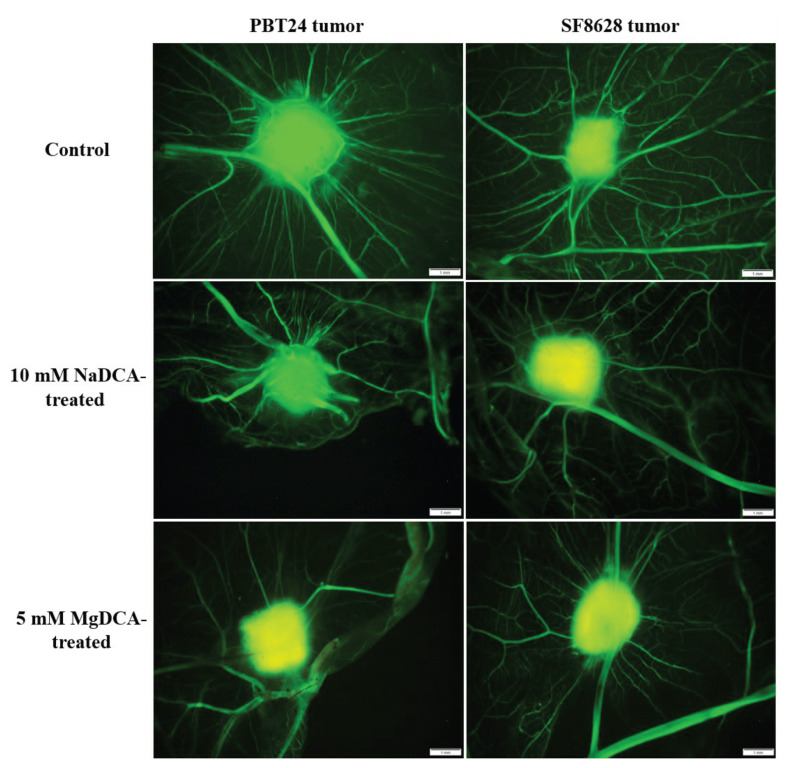
Fluorescent stereomicroscopic images of PBT24 and SF8628 tumors (fluorescent dextran highlights the tumor and surrounding blood vessels). Scale bar is 1 mm.

**Figure 3 ijms-23-10455-f003:**
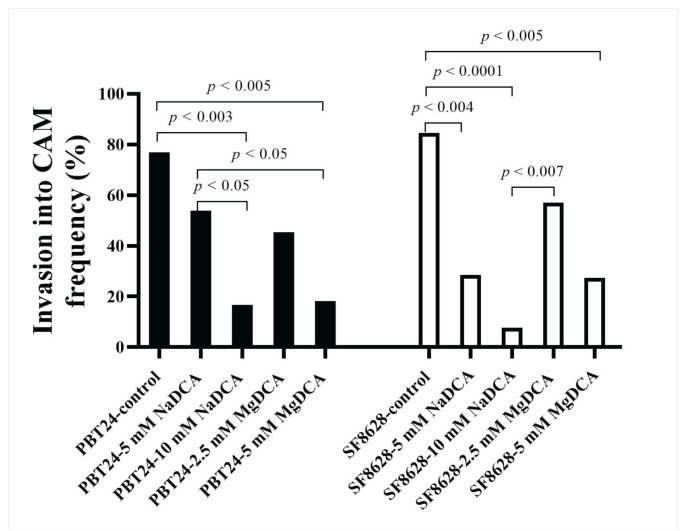
The invasion into CAM frequency in the PBT24 and SF8628 tumor groups.

**Figure 4 ijms-23-10455-f004:**
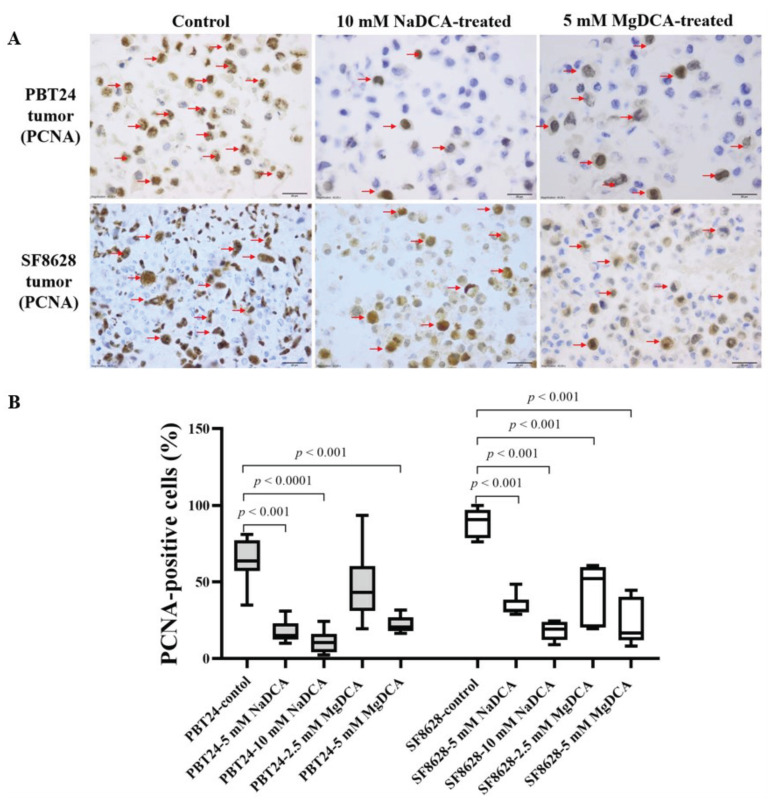
Immunohistochemical PCNA expression of PBT24 and SF8628 in control and DCA-treated tumor groups. (**A**) Dark brown nuclei indicate PCNA-positive cells (red arrows). The magnification is 40×, scale bar is 20 µm. (**B**) Percentage of PCNA-positive cells in the tissue of PBT24 and SF8628 tumors.

**Figure 5 ijms-23-10455-f005:**
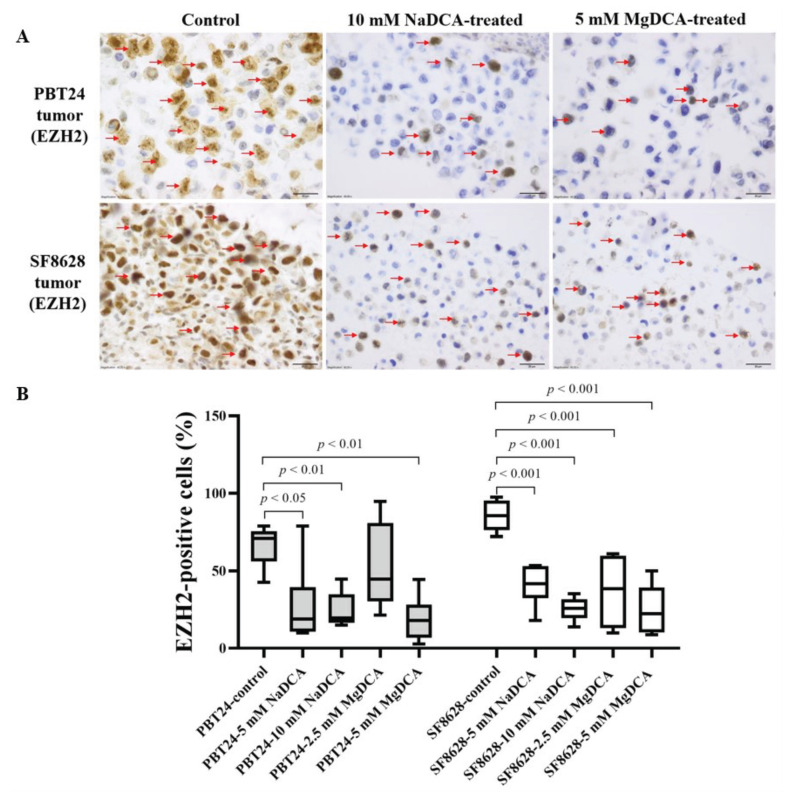
Immunohistochemical EZH2 expression in control and treated tumor groups. (**A**) Dark brown nuclei indicate EZH2-positive cells (red arrows). The magnification is 40×, scale bar is 20 µm. (**B**) The percentage of EZH2-positive cells in PBT24 and SF8628 tumors.

**Figure 6 ijms-23-10455-f006:**
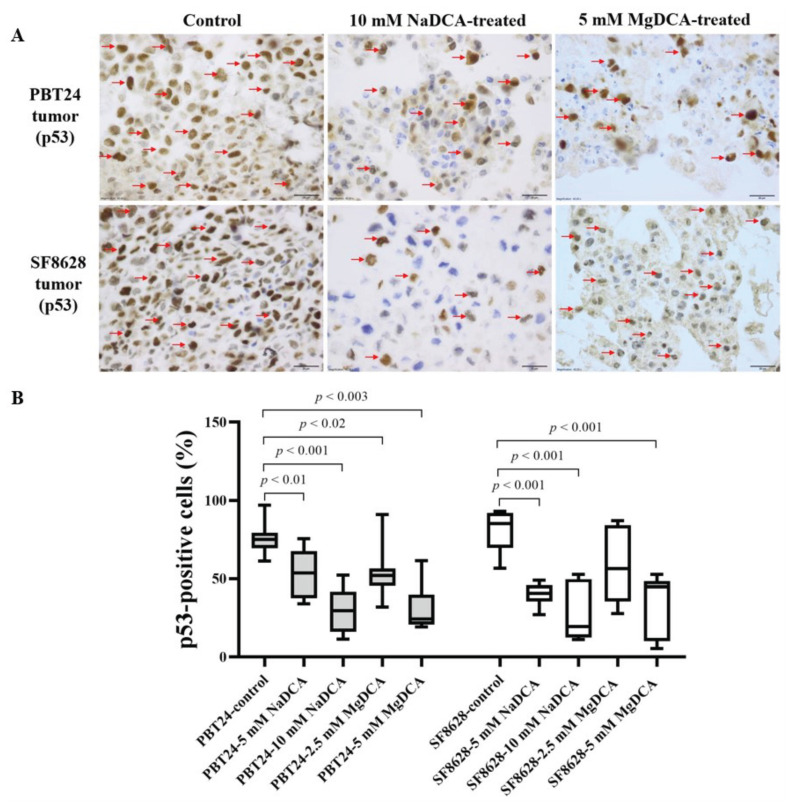
Immunohistochemical expression of p53 in control and DCA-treated tumor groups. (**A**) Dark brown nuclei indicate p53-positive cells (red arrows). The magnification is 40×, scale bar is 20 µm. (**B**) The percentage of p53-positive cells in PBT24 and SF8628 tumors.

**Figure 7 ijms-23-10455-f007:**
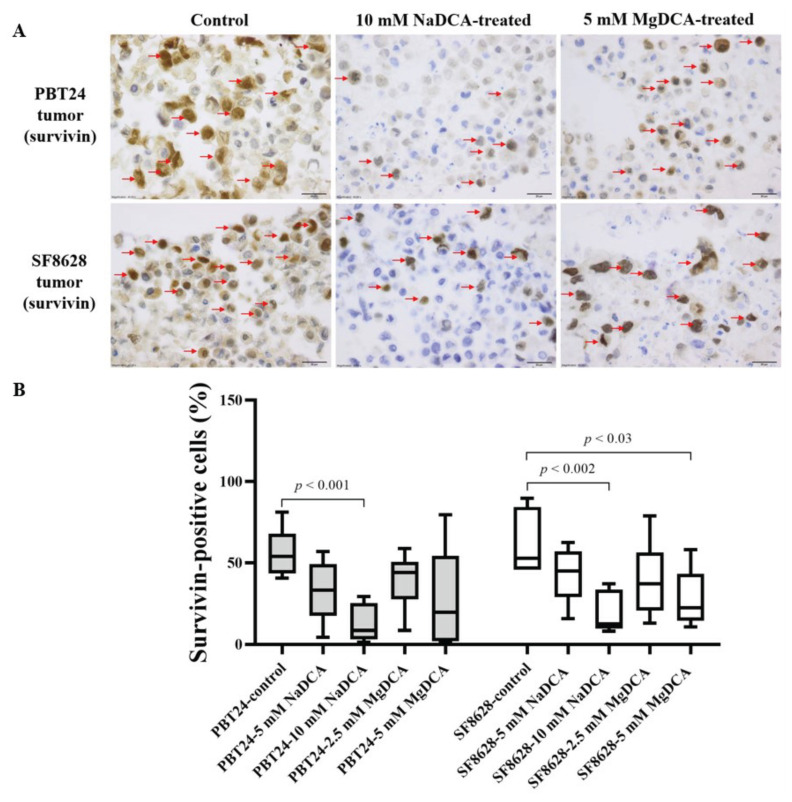
Immunohistochemical expression of survivin in control and DCA-treated PBT24 and SF8628 tumors. (**A**) Dark brown nuclei indicate survivin-positive cells (red arrows). The magnification is 40×, scale bar is 20 µm. (**B**) The percentage of survivin-positive cells in PBT24 and SF8628 tumors.

**Figure 8 ijms-23-10455-f008:**
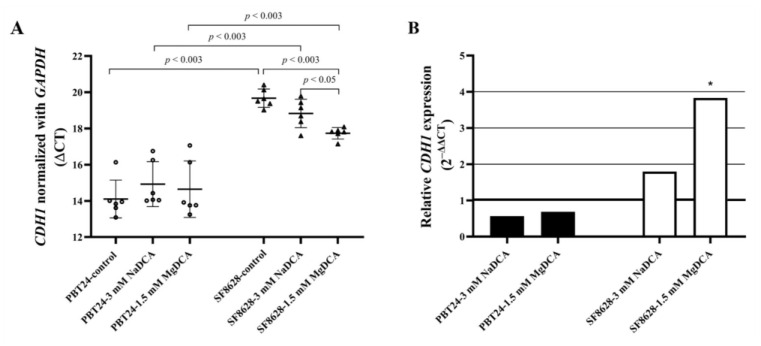
*CDH1* expression in PBT24 and SF8628 control and dichloroacetate-treated cell groups. (**A**) The graph shows the ΔCT value, it’s mean (long horizontal bar), and SD value (short horizontal bars). (**B**) Relative *CDH1* expression of PBT24 and SF8628 cell groups treated with dichloroacetate preparations. *CDH1* expression of the treated group was compared with the corresponding control. The 1.0 line denotes the reference value of gene expression; * *p* < 0.05.

**Table 1 ijms-23-10455-t001:** The invasion into CAM frequency, the number of a blood vessels in CAM under the tumor, and the CAM thickness data of the PBT24 and SF8628 tumors.

Study Group	*n*	Invasion Frequency (%)	Number of Blood Vessels	CAM Thickness (µm)
			Median (Range)	
PBT24-control	13	76.9	15 (6–28)	300.9 (65.2–700.9)
PBT24-5 mM NaDCA	13	53.9 ^a,b^	12 (5–29)	236.02 (31.3–484.4)
PBT24-10 mM NaDCA	12	16.7 ^c^	11.5 (3–25)	199.2 (54.4–627.6)
PBT24-2.5 mM MgDCA	11	45.5	9 (2–23)	138.6 (21.3–383.0) ^n^
PBT24-5 mM MgDCA	11	18.2 ^d^	7 (2–27) ^i^	96.5 (36.3–591.9) ^o^
SF8628-control	13	84.6	15 (5–21)	282.5 (47.85–539.7)
SF8628-5 mM NaDCA	14	28.6 ^e^	6 (1–22) ^j^	247.5 (86.76–879.3)
SF8628-10 mM NaDCA	13	7.7 ^f^	5 (2–16) ^k^	199.4 (52.09–776.2)
SF8628-2.5 mM MgDCA	14	57.1 ^g^	10.5 (5–23) ^l^	244.2 (55.02–409.1)
SF8628-5 mM MgDCA	11	27.3 ^h^	8 (2–31) ^m^	140.5 (50.56–636.4)

^a^ *p* < 0.05, compared with the PBT24-10 mM NaDCA; ^b^ *p* < 0.05, compared with the PBT24-5 mM MgDCA; ^c^
*p* < 0.003, compared with the PBT24-control; ^d^
*p* < 0.005, compared with the PBT24-control; ^e^
*p* < 0.004, compared with the SF8628-control; ^f^
*p* < 0.0001, compared with the SF8628-control; ^g^
*p* < 0.007, compared with the SF8628-10 mM NaDCA; ^h^
*p* < 0.005, compared with the SF8628-control; ^i^
*p* < 0.02, compared with the PBT24-control; ^j^
*p* < 0.04, compared with the SF8628-control; ^k^
*p* < 0.001, compared with the SF8628-control; ^l^
*p* < 0.002, compared with the SF8628-10 mM NaDCA; ^m^ < 0.03, compared with the SF8628-10 mM NaDCA; ^n^
*p* < 0.05, compared with the PBT24-control; ^o^
*p* < 0.02, compared with the PBT24-control.

**Table 2 ijms-23-10455-t002:** The PCNA-positive cells percentage in PBT24 and SF8628 tumors.

Study Group	PBT24 Tumor PCNA-Positive Cells (%)		SF8628 Tumor PCNA-Positive Cells (%)	
	*n*	Median (Range)	*n*	Median (Range)
Control	9	63.8 (34.9–81.0)	8	90.8 (76.3–100)
5 mM NaDCA	6	15.0 (10.0–31.1) ^a,b^	7	31.5 (29.0–48.5) ^g,h^
10 mM NaDCA	8	10.5 (2.4–24.3) ^c,d^	7	19.1 (9.1–24.6) ^i^
2.5 mM MgDCA	7	43.1 (19.5–93.6)	6	52.2 (19.3–60.7) ^j,k,l^
5 mM MgDCA	6	20.7 (16.4–31.7) ^e,f^	7	16.7 (8.2–44.6) ^m^

^a^ *p* < 0.001, compared with the PBT24-control; ^b^ *p* < 0.005, compared with the PBT24-2.5 mM MgDCA; ^c^
*p* < 0.0001, compared with the PBT24-control; ^d^ *p* < 0.001, compared with the PBT24-2.5 mM MgDCA; ^e^
*p* < 0.001, compared with the PBT24-control; ^f^ *p* < 0.03, compared with the PBT24-2.5 mM MgDCA; ^g^
*p* < 0.001, compared with the SF8628-control; ^h^ *p* < 0.001, compared with the SF8628-10 mM NaDCA; ^i^
*p* < 0.001, compared with the SF8628-control; ^j^
*p* < 0.001, compared with the SF8628-control; ^k^ *p* < 0.04, compared with the SF8628-10 mM NaDCA; ^l^ *p* < 0.04, compared with the SF8628-5 mM MgDCA; ^m^
*p* < 0.001, compared with the SF8628-control.

**Table 3 ijms-23-10455-t003:** The EZH2-positive cells percentage in PBT24 and SF8628 tumors groups.

Study Group	PBT24 Tumor EZH2-Positive Cells (%)		SF8628 Tumor EZH2-Positive Cells (%)	
	*n*	Median (Range)	*n*	Median (Range)
Control	6	71.0 (42.6–78.7)	8	85.4 (72.0–97.5)
5 mM NaDCA	6	18.8 (9.8–78.7) ^a,b^	6	41.6 (18.0–53.5) ^g,h^
10 mM NaDCA	7	19.6 (14.9–44.6) ^c,d^	7	25.9 (13.8–35.3) ^i^
2.5 mM MgDCA	7	44.7 (21.5–94.8)	6	38.4 (9.9–61.1) ^j^
5 mM MgDCA	7	17.9 (2.8–44.4) ^e,f^	6	22.4 (8.9–50.0) ^k^

^a^*p* < 0.05, compared with the PBT24-control; ^b^
*p* < 0.03, compared with the PBT24-2.5 mM MgDCA; ^c^ *p* < 0.01, compared with the PBT24-control; ^d^ *p* < 0.03, compared with the PBT24-2.5 mM MgDCA; ^e^ *p* < 0.01, compared with the PBT24-control; ^f^ *p* < 0.01, compared with the PBT24-2.5 mM MgDCA; ^g^
*p* < 0.001, compared with the SF8628-control; ^h^
*p* < 0.04, compared with the SF8628-10 mM NaDCA; ^i^
*p* < 0.001, compared with the SF8628-control; ^j^
*p* < 0.001, compared with the SF8628-control; ^k^
*p* < 0.001, compared with the SF8628-control.

**Table 4 ijms-23-10455-t004:** The p53-positive cells percentage in PBT24 and SF8628 tumors groups.

Study Group	PBT24 Tumor p53-Positive Cells (%)		SF8628 Tumor p53-Positive Cells (%)	
	*n*	Median (Range)	*n*	Median (Range)
Control	7	75.1 (61.2–96.9)	8	85.3 (56.7–92.9)
5 mM NaDCA	8	53.7 (33.9–75.5) ^a^	6	40.7 (27.1–48.9) ^h^
10 mM NaDCA	8	29.6 (11.5–52.3) ^b,c,d^	6	19.5 (11.3–52.6) ^i^
2.5 mM MgDCA	7	51.9 (32.0–91.0) ^e^	6	56.3 (27.8–87.1) ^j^
5 mM MgDCA	6	24.2 (19.3–61.6) ^f,g^	7	44.9 (5.4–52.8) ^k^

^a^*p* < 0.01, compared with the PBT24-control; ^b^
*p* < 0.001, compared with the PBT24-control; ^c^
*p* < 0.02, compared with the PBT24-5 mM NaDCA; ^d^
*p* < 0.01, compared with the PBT24-2.5 mM MgDCA; ^e^
*p* < 0.02, compared with the PBT24-control; ^f^
*p* < 0.003, compared with the PBT24-control; ^g^
*p* < 0.03, compared with the PBT24-5 mM NaDCA; ^h^
*p* < 0.001, compared with the SF8628-control; ^i^
*p* < 0.001, compared with the SF8628-control; ^j^
*p* < 0.05, compared with the SF8628-10 mM NaDCA; ^k^
*p* < 0.001, compared with the SF8628-control.

**Table 5 ijms-23-10455-t005:** The percentage of survivin-positive cells in PBT24 and SF8628 tumors groups.

Study Group	PBT24 Tumor Survivin- Positive Cells (%)		SF8628 Tumor Survivin-Positive Cells (%)	
	*n*	Median (Range)	*n*	Median (Range)
Control	6	53.8 (40.6–81.1)	6	52.8 (45.7–89.6)
5 mM NaDCA	6	33.1 (4.3–56.8)	7	45.0 (15.6–62.3)
10 mM NaDCA	8	8.5 (1.3–29.3) ^a,b,c^	7	12.4 (8.1–37.0) ^d,e^
2.5 mM MgDCA	7	43.9 (8.4–58.7)	6	37.0 (13.0–8.6)
5 mM MgDCA	6	19.6 (0.9–79.4)	6	22.4 (10.6–58.1) ^f^

^a^*p* < 0.001, compared with the PBT24-control; ^b^
*p* < 0.05, compared with the PBT24-5 mM NaDCA; ^c^
*p* < 0.006, compared with the PBT24-2.5 mM MgDCA; ^d^
*p* < 0.02, compared with the SF8628-5 mM NaDCA; ^e^
*p* < 0.002, compared with the SF8628-control; ^f^
*p* < 0.03, compared with the SF8628-control.

**Table 6 ijms-23-10455-t006:** The expression of *SLC12A2*, *SLC12A5*, *SLC5A8*, and *GAPDH* genes in tested PBT24 and SF8628 cells.

Study Group	*n*	CT Mean		ΔCT Mean ± SD	ΔΔCT
		*SLC12A2*	*GAPDH*		
PBT24-control	6	22.95	19.37	3.58 ± 0.73	
PBT24-3 mM NaDCA	6	22.72	18.69	4.03 ± 0.48	0.45
PBT24-1.5 mM MgDCA	6	23.00	19.31	3.69 ± 1.33	0.11
SF8628-control	6	22.89	19.02	3.88 ± 0.21	
SF8628-3 mM NaDCA	6	23.61	19.77	3.84 ± 0.29	−0.04
SF8628-1.5 mM MgDCA	6	23.66	19.63	4.04 ± 0.27	0.16
		* **SLC12A5** *	* **GAPDH** *	**ΔCT Mean ± SD**	**ΔΔCT**
PBT24-control	6	32.56	19.37	13.19 ± 0.83	
PBT24-3 mM NaDCA	6	31.94	18.69	13.25 ± 0.72	0.06
PBT24-1.5 mM MgDCA	6	32.39	19.31	13.08 ± 1.19	−0.11
SF8628-control	6	36.83	19.02	17.81 ± 0.43 ^a^	
SF8628-3 mM NaDCA	6	37.84	19.77	18.07 ± 0.81 ^b^	0.260
SF8628-1.5 mM MgDCA	6	37.66	19.63	18.04 ± 0.29 ^c^	0.223
		* **SLC5A8** *	* **GAPDH** *	**ΔCT Mean ± SD**	**ΔΔCT**
PBT24-control	6	39.42	19.37	20.04 ± 1.82	
PBT24-3 mM NaDCA	6	38.91	18.69	20.22 ± 1.81	0.18
PBT24-1.5 mM MgDCA	6	39.47	19.31	20.17 ± 2.24	0.12
SF8628-control	6	38.97	19.02	19.96 ± 0.59	
SF8628-3 mM NaDCA	6	40.06	19.77	20.29 ± 0.68	0.34
SF8628-1.5 mM MgDCA	6	40.03	19.63	20.41 ± 0.54	0.45

^a^*p* < 0.003, compared with PBT24-control; ^b^
*p* < 0.003, compared with PBT24-3 mM NaDCA; ^c^
*p* < 0.003, compared with PBT24-1.5 mM MgDCA.

**Table 7 ijms-23-10455-t007:** The *CDH1, CDH2*, and *GAPDH* expression in the studied PBT24 and SF8628 cell groups.

Study Group	*n*	CT Mean		ΔCT Mean ± SD	ΔΔCT
		*CDH1*	*GAPDH*		
PBT24-control	6	33.48	19.37	14.104 ± 1.1	
PBT24-3 mM NaDCA	6	33.61	18.69	14.92 ± 1.24	0.82
PBT24-1.5 mM MgDCA	6	33.95	19.31	14.65 ± 1.56	0.54
SF8628-control	6	38.69	19.02	19.67 ± 0.51 ^a^	
SF8628-3 mM NaDCA	6	38.59	19.77	18.83 ± 0.79 ^b^	−0.84
SF8628-1.5 mM MgDCA	6	37.36	19.63	17.73 ± 0.32 ^c,d,e^	−1.94
		* **CDH2** *	* **GAPDH** *	**ΔCT Mean ± SD**	**ΔΔCT**
PBT24-control	6	22.92	19.37	3.55 ± 0.98	
PBT24-3 mM NaDCA	6	22.55	18.69	3.86 ± 0.85	0.31
PBT24-1.5 mM MgDCA	6	22.28	19.31	2.98 ± 1.09	−0.57
SF8628-control	6	23.45	19.02	4.43 ± 0.23	
SF8628-3 mM NaDCA	6	23.95	19.77	4.18 ± 0.39	−0.25
SF8628-1.5 mM MgDCA	6	24.14	19.63	4.52 ± 0.19 ^f^	0.08

^a^*p* < 0.003, compared with the PBT24-control; ^b^
*p* < 0.003, compared with PBT24-3 mM NaDCA; ^c^
*p* < 0.003, compared with the SF8628-control; ^d^
*p* < 0.05, compared with the SF8628-3 mM NaDCA; ^e^
*p* < 0.003, compared with the PBT24-1.5 mM MgDCA; ^f^
*p* < 0.03, compared with the PBT24-1.5 mM MgDCA.

**Table 8 ijms-23-10455-t008:** Control and treated PBT24 and SF8628 tumors on CAM groups, tumor markers expression study groups, and sample size.

Control and Treated Group	Invasion, No. of Vessels, CAM Thickness		PCNA		EZH2		p53		Survivin	
	PBT24 *n*	SF8628 *n*	PBT24 *n*	SF8628 *n*	PBT24 *n*	SF8628 *n*	PBT24 *n*	SF8628 *n*	PBT24 *n*	SF8628 *n*
Control	13	13	9	8	6	8	7	8	6	6
5 mM NaDCA	13	14	6	7	6	6	8	6	6	7
10 mM NaDCA	12	13	8	7	7	7	8	6	8	7
2.5 mM MgDCA	11	14	7	6	7	6	7	6	7	6
5 mM MgDCA	11	11	6	7	7	6	6	7	6	6

## Data Availability

The data presented in this study are available on request from the corresponding author.

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
