# Peer review of "The Comparative Experimental Study of Sodium and Magnesium Dichloroacetate Effects on Pediatric PBT24 and SF8628 Cell Glioblastoma Tumors Using a Chicken Embryo Chorioallantoic Membrane Model and on Cells In Vitro"

_ijms, 2022, doi:10.3390/ijms231810455_

Round 1

Reviewer 1 Report

The MS presents results of experiments on high-grade pediatric glioblastoma (phGBM) cell lines carried out with the use of chorioallantoic membrane (CAM) model for tumor xenografts as well as classical adherent cell cultures. PBT-24 and SF8628 cell lines were utilized to evaluate the antiproliferative and anti-angiogenic effects of different dichloroacetate (DCA) preparations including its sodium or magnesium salts. Authors tested the effects of DCA on the invasion, angiogenesis and immunoexpression of markers associated with the progression of phGBM in tumor xenografts utilizing the CAM model. The effects of DCA treatment on expression levels of SLC12A5, SLC12A2 and SCL5A8 co-transporters and cadherins were investigated in adherent cultures of PBT24 and SF8628 cells by QPCR. The submitted MS continues the series of articles on GBM recently published by the authors . The present study provide novel, perhaps clinically significant insights on the treatment modalities in pediatric GBM, suggesting the direction for further experiments and trials.  Presented study was well planned, the data are presented in a proper form and the results support the conclusions drawn by the authors. However, there are some issues that would improve the quality of MS if they were addressed by the authors in the revised version of submitted MS.

Major remarks:

1.       Introduction is too long and difficult to follow. It includes 38 citations and some paragraphs of this chapter already provide some discussion rather than direct to the aims of submitted study. Authors shall consider to revise the introduction by shortening it and transferring/merging the redundant sentences/paragraphs (e.g. adverse effects and administration of DCA and some others) with the respective sections of the Discussion chapter.

2.       In the Results lines 122-126 are redundant because they repeat the legend for the Figure 1.

3.       What is the purpose of presenting statistical differences between the 2.5 MgDCA and 10.0 NaDCA treatments (e.g. Table 4, Figure 6) since these treatments differ by both, anion concentration and cation type.

4.       There is no need to repeat every Ct, dCt and ddCt value for QPCR in the text (subsection 2.7) while they are clearly presented in the tables (Table 6 and 7) and graph (Figure 8).

5.       The subsection titles within the discussion chapter are not recommended.

6.       The interpretation and discussion of results provided by CAM model-based experiments and gene expression analysis in cell cultures shall be followed by a more synthetic paragraph summarizing presented findings and focusing more on the novelty of the current study. Possible outcomes coming from genetic banckground/sex differences between the cell lines shall be addressed better. Authors shall underlie the progress they made in comparison to their previous studies utilizing CAM model and GBM tumors/cell cultures as well as summarize clinical significance of their findings.

7.       The order of references is incorrect: 1-38 in the Introduction, then 45-96 in the Discussion and 39-44 in the Methods.

8.       Information about primary antibody against p53 is missing (subsection 4.4, page 21). In addition I would suggest t provide in the text basic information about all primary antibody used in the study (host, type, catalogue number and/or clone ID).

Minor remarks:

9.       Line 521 – capitalize “Ezh2”, remove italicization.

10.   Line 547-551 – it needs to be made precise that down-regulation/inhibition of p53 is considered beneficial due to overexpression of its mutant/oncogenic but not wild types.

11.    Lines 578-584 – what is the purpose/usefulness for the treatment of GBM with the agent that up-regulates SLC12A2?

12.   Subsection 4.2 authors shall consider to present experimental groups in the table (or tables).

Author Response

Dear Reviewer,

Thank you very much for your remarks and your positive evaluation of the manuscript. We are sending you the replies to your comments and an explanation of how the manuscript has been revised in the light of the comments received. We believe that the corrections made have certainly improved the manuscript and we express our appreciation for your help.

Major remarks:

  1. Introduction is too long and difficult to follow. It includes 38 citations and some paragraphs of this chapter already provide some discussion rather than direct to the aims of submitted study. Authors shall consider to revise the introduction by shortening it and transferring/merging the redundant sentences/paragraphs (e.g. adverse effects and administration of DCA and some others) with the respective sections of the Discussion chapter.

Answer

We have reduced the number of references in line with the remark: references 2, 6, 12, 29, 30, 31 have been removed from the original version, and references 23, 24 and 26 have been moved to the Discussion. As a result, the Introduction section has become shorter and smoother, and the number of references in the Introduction has been reduced by nine. Thanks for the remark.

  1. In the Results lines 122-126 are redundant because they repeat the legend for the Figure 1.

Answer

The text (lines 122-126) has been removed. Thank you for your note.

  1. What is the purpose of presenting statistical differences between the 2.5 MgDCA and 10.0 NaDCA treatments (e.g. Table 4, Figure 6) since these treatments differ by both, anion concentration and cation type.

Answer

This study is a comparative study assessing the differences in efficacy between NaDCA and MgDCA salts, which have the same active substance (dichloroacetate anion). Therefore, we think that showing the differences is also meaningful when comparing the effects of different salts doses. We would therefore like to keep in the text the comparisons on which you commented. We have therefore not made any corrections in line with the comment.

  1. There is no need to repeat every Ct, dCt and ddCt value for QPCR in the text (subsection 2.7) while they are clearly presented in the tables (Table 6 and 7) and graph (Figure 8).

Answer

Corrections made in line with the comment. Thank you very much for it.

  1. The subsection titles within the discussion chapter are not recommended.

Answer

The titles of the subsections of the Discussion section have been removed. Thank you for your note.

  1. The interpretation and discussion of results provided by CAM model-based experiments and gene expression analysis in cell cultures shall be followed by a more synthetic paragraph summarizing presented findings and focusing more on the novelty of the current study. Possible outcomes coming from genetic banckground/sex differences between the cell lines shall be addressed better. Authors shall underlie the progress they made in comparison to their previous studies utilizing CAM model and GBM tumors/cell cultures as well as summarize clinical significance of their findings.

Answer

We think that by making additions to the Discussion in response to the first comment, we have fulfilled the requirement by partially moving the text into the Discussion from the Introduction. We would not like to expand further as the Discussion is already quite extensive. Thank you for your comment, it will be useful to us in the preparation of future publications.

  1. The order of references is incorrect: 1-38 in the Introduction, then 45-96 in the Discussion and 39-44 in the Methods.

Answer

Thank you for your important comment. The inaccuracies have been corrected as requested.

  1. Information about primary antibody against p53 is missing (subsection 4.4, page 21). In addition I would suggest t provide in the text basic information about all primary antibody used in the study (host, type, catalogue number and/or clone ID).

Answer

The inaccuracies have been corrected as requested.

Minor remarks:

  1. Line 521 – capitalize “Ezh2”, remove italicization.

Answer

Comment fulfilled

  1. Line 547-551 – it needs to be made precise that down-regulation/inhibition of p53 is considered beneficial due to overexpression of its mutant/oncogenic but not wild types.

Answer

Thank you for your advice. The text has been updated accordingly.

  1. Lines 578-584 – what is the purpose/usefulness for the treatment of GBM with the agent that up-regulates SLC12A2?

Answer

Thank you for your comment. The usefulness of treating GBM with a product that activates SLC12A2 requires research. We are also thinking that such an effect of an agent (e.g. temozolomide) may be the reason for the failure of treatment. However, in the context of individual therapy, it is necessary to investigate the effect on the activity of other markers that regulate intracellular chloride levels. A more robust judgement can then be made about the potential risks. Our study, which partially answers the questions in the context of individual therapy, is aimed at investigating such mechanisms. This is very important for deciding on drug combination therapy.

  1. Subsection 4.2 authors shall consider to present experimental groups in the table (or tables).

Answer

The requirement is fulfilled.

Remark

(x) English language and style are fine/minor spell check required

Answer

English language of the manuscript was revised and corrected.

Again, we appreciate your comments. We believe that by making corrections in line with them we have really improved the quality of the manuscript.

Reviewer 2 Report

Dear Authors,

well written manuscript about the promising for future treatment pathogenesis in pediatric glioblastoma, thank you. I have just some small remarks regarding the manuscript:

1) would be great to decipher abbreviations behind the titles of Figs; also indicate please that the Figs. 4-7 represent the IHC micrographs in the title of each Fig or after it;

2) please, add the Limitation paragraph at the end of Discussion;

3) References part is very excellent! Are you sure that there is no possible to remove or exchange 2 old previous century literature sources what simply do not fit for this very nice manuscript?

Author Response

Dear Reviewer,

Thank you very much for your comments and positive evaluation of the manuscript. We are sending you the responses to your comments and an explanation of how the manuscript has been revised in the light of the comments received.

1) would be great to decipher abbreviations behind the titles of Figs; also indicate please that the Figs. 4-7 represent the IHC micrographs in the title of each Fig or after it;

Answer

We have made the corrections in line with the comment.

2) please, add the Limitation paragraph at the end of Discussion;

Answer

We have made the corrections in line with the comment.

3) References part is very excellent! Are you sure that there is no possible to remove or exchange 2 old previous century literature sources what simply do not fit for this very nice manuscript?

Answer

We have made the corrections in line with the comment. We have removed reference 12 from the manuscript (original version).

Remark

(x) English language and style are fine/minor spell check required

Answer

English language of the manuscript was revised and corrected.

 We believe that the corrections made have definitely improved the manuscript. Again, thank you for your comments, which have helped to improve the manuscript.
